# Exploring the science and data foundation for Federal public lands decisions

**Alison C. Foster**[1¤☯*], **Andrew T. Canchola**[2☯], **Travis S. Haby**[3], **Sarah K. Carter**[1]

**1** U.S. Geological Survey Fort Collins Science Center, Fort Collins, Colorado, United States of America, **2** Contractor with the U.S. Geological Survey, Fort Collins Science Center, Fort Collins, Colorado, United States of America, **3** Bureau of Land Management, National Operations Center, Denver, Colorado, United States of America

☯ These authors contributed equally to this work.
¤ Current address: USDA Forest Service Rocky Mountain Regional Office, Lakewood, Colorado, United States of America
* alison.c.foster@gmail.com

## Abstract

Public lands provide diverse resources, values, and services worldwide. Laws and policies typically require consideration of science in public lands decisions, and resource managers are committed to science-informed decision-making. However, it can be challenging for managers to use, and document the use of, science and data in their decisions. To better understand science and data use in Federal public lands decisions in the United States, we assessed the number, type, and age of documents cited in 70 Environmental Assessments (EAs) completed by the Bureau of Land Management (BLM) in Colorado from 2015–2019. We focused on the BLM, as they manage the largest area of public lands in the United States. We selected Colorado as our study area, as actions proposed on BLM lands in Colorado are representative of those across the nation. Fifty percent of citations were categorized as science and 23% as data. EAs contained an average of 17 citations (range 0–111), with documents analyzing effects of oil and gas development and recreation actions including the highest and lowest mean number of citations (41 and 6, respectively). Of individual resource analysis sections within EAs, 24% contained ≥1 science citation and 21% contained ≥1 data citation. Journal articles were the most cited type of document (26% of citations) followed by non-BLM inventories (13%). Forty-seven percent of citations were relatively recent (2010 or later); the oldest citation was from 1927. Commonly analyzed resources with the highest mean number of citations were socio-economics, mineral resources, and noise. Fourteen of 33 commonly analyzed resources included <1 citation on average. Actions and resources with no or few citations represent opportunities for strengthening the transparent use of science and data in public lands decision-making.

## Introduction

Public lands are important in the United States and worldwide for many reasons, including for extractive uses such as timber harvest and energy production along with conservation of

**Data availability statement:** Data is available via the U.S. Geological Survey data repository. https://doi.org/10.5066/P142RPVE

**Funding:** Dr. Sarah Carter received funding for this project from the U.S. Bureau of Land Management (BLM) National Operations Center under agreements L16PG00147 and L21PG00092. None of the funders played a role in study design, data collection and analysis, decision to publish, or preparation of the manuscript. Information about the BLM National Operations Center can be found at https://www.blm.gov/services/national-operations-center.

**Competing interests:** The authors have declared that no competing interests exist.

rare species, recreation, and therapeutic opportunities [1–3]. Public lands provide valuable ecosystem services such as clean air and water [4], and proximity to public lands can stimulate local economies [3]. Public lands also provide important intrinsic, aesthetic, and cultural values, and people build strong attachment to their public lands [1,5]. Additionally, public lands will be essential to meeting recent land conservation targets and climate change mitigation goals [6]. Management of public lands can thus be challenging, as land managers must balance many competing interests, uses, and values in their decisions.

Environmental laws and policies in the United States and in many countries around the world typically require that major decisions affecting public lands and resources consider environmental effects and that they use science and data in the decision-making process [7]. Major decisions on all Federal public lands in the United States that may affect the quality of the environment are subject to review under the National Environmental Policy Act (NEPA, 42 U.S.C. § 4321). NEPA requires use of the natural and social sciences in decisions (42 U.S.C. § 4322(2)(A)). The U.S. Endangered Species Act also requires science use: specifically, that species status determinations be made on the basis of the 'best scientific and commercial data available' (16 U.S.C. §1533(b)(1)(A)). The U.S. presidential (Joseph Biden) administration has also emphasized the need to use the best available science to guide decision-making [8], and U.S. Federal agencies are committed to science-informed management (36 C.F.R. §219.3, [9]).

Resource managers are interested in and committed to science-informed decision-making [10,11], and specifically to integrating the best available science into the NEPA process [12]. Studies of landscape-level planning have also suggested that greater integration of landscape science into plans may benefit public land management [13]. Strong use of science and data in decision-making may help reduce the chances of litigation, as prior studies have shown that legal challenges to public land decision-making include allegations about a lack of science and data use in decisions [14].

The characteristics of science and data used in decision-making are important. NEPA focuses on use of relevant, credible, and reliable data and science (40 CFR §§1500.1, 1502.21(d), 1502.23), while the Endangered Species Act refers to the best available science and commercial data (16 U.S.C. §1533(b)(1)(A)). A number of studies have explored definitions for and characteristics of what is commonly referred to as best available science information. Accuracy, reliability, credibility, and relevancy are core characteristics of the best available science [15]. Independent, peer review of research methods, results, and conclusions – key aspects of the publication process – are considered fundamental for science credibility (e.g., [15,16]). Empirical research conducted using the scientific method is considered the highest standard (e.g., [17,18]). While different groups can have different perceptions of what is most important in characterizing best available science, the quality of the methodology often emerges as a top characteristic [19]. The institution or organization producing the science can also affect its credibility, with public and private research universities and the U.S. Geological Survey (a Federal science provider) being perceived as most credible [19]. Finally, the scientific process tests assumptions and addresses uncertainties, limitations, and inconsistencies over time, meaning that how recently the science was conducted and published, along with its rigor, is also relevant to whether it constitutes the best available science [20].

Despite requirements for, and commitments to, the use of science in public lands decision-making, there are multiple factors that make it challenging. Management agencies and their staff may lack time and capacity for finding, reading, synthesizing, and applying science to their decisions [15,17,21], particularly for agencies with chronic staff shortages and high staff turnover rates. Access to scientific literature remains a barrier [11,22], along with science that

may not be presented in a format or conducted at a scale that is most relevant and useable for resource managers [23,24]. Additionally, relevant science may be readily available for some topics, such as wildland fire management [25], but may not be as available or accessible for topics that are emerging priorities or topics for which peer-reviewed scientific studies are generally less available (e.g., visual or scenic resources). There can also be a disconnect between the topics on which researchers publish (e.g., [26]) and the topics that are the highest priorities for resource managers [27]. Potential biases and pressures, including political pressures, on decision-making processes can also make use of the best available science challenging [17]. These and other barriers to integrating science into resource management have been well-studied [21,28–30].

Natural resource management and conservation decisions may also be informed by a range of information sources in addition to science, including statutes, regulations, planning documents, professional experience, data, and technical information [31,32]. Resource managers also often rely heavily on personal experience or professional judgement to inform their decision-making [30,33,34]. For example, [33] found that 90% of resource assessments are made using experience-based knowledge, and [30] found that resource managers used expert opinion more often than science to support their decisions in management plans.

Given this setting of legal and policy requirements for science to be used in public lands decisions, known challenges to that use, and frequent reliance on other types of information to inform decisions, we sought to assess the current status of clear science use in public lands decision-making. Specifically, we sought to identify the types of information/documents, including science and data products (hereafter collectively referred to as documents), that resource managers of Federal public lands are currently using and citing in their decision-making. We focused our study on the Bureau of Land Management (BLM), which manages approximately 1/10th of the nation's surface acres, more than any other public land management agency in the United States, and on the NEPA analyses that the agency conducts to assess the potential environmental effects of its decisions. Our objectives were to characterize the number, age, and document type of citations in BLM NEPA analyses 1) across the state of Colorado, 2) by type of project or proposed action (e.g., energy development, permitting livestock grazing), and 3) by potentially affected resources that were analyzed in decisions (e.g., rare plants, protected birds). Understanding current science use is an important first step to supporting greater science integration into public lands decision-making. We suggest that understanding the types of science and data documents cited in BLM NEPA analyses may assist both management agencies and science providers in addressing potential barriers to science integration.

## Methods

We chose the BLM as our study agency because they manage the largest area of public lands in the United States. The BLM's mandate under the Federal Land Policy and Management Act of 1976 requires managing public lands for multiple resources, uses, and values and for sustained yield of renewable resources (43 U.S.C. §§ 1701–1787), which can be nuanced and challenging [35].

We selected Colorado as our study area because we had established relationships with BLM planners, managers, and policy makers in the state which allowed us to work within a coproduction framework [36,37]. Working together as partners allowed us to jointly develop study methods and regularly share findings with decision-makers. BLM-managed lands in Colorado also span a range of ecosystem types (including high deserts, forests and woodlands, riparian areas, and alpine zones), and the frequency distribution of different types of actions (e.g., energy development, livestock grazing) that are proposed on BLM-managed lands in Colorado is similar to that across the nation during the same time period [27].

We analyzed citations in NEPA analyses conducted to inform public lands decisions made by the BLM in Colorado to understand the characteristics of science and data cited in these documents. We chose to focus on Environmental Assessments (EAs) because EAs constitute the vast majority of NEPA analyses conducted by the BLM each year to assess the potential effects of proposed management decisions and actions (e.g., a newly proposed recreation trail) on resources (e.g., wildlife habitat) and existing resource uses (e.g., current recreation activities and user experiences) of concern. For simplicity, we refer to these resources and existing resource uses hereafter as just resources.

To identify our study sample of EAs, we downloaded a list of all EAs available on the BLM's National NEPA Register (https://www.blm.gov/programs/planning-and-nepa/eplanning) for the years 2015–2019. We considered only those EAs categorized on the website as complete (i.e., project status listed as completed, decision and protest, or decision and appeal) to ensure we were looking at final documents. We then selected a stratified random sample of BLM EAs as follows: we sampled 10 EAs from BLM-assigned categories of proposed actions for which there were at least 10 completed EAs in Colorado (Fish and wildlife, Fluid minerals [hereafter, Oil and gas], Lands and realty [includes land transactions as well as rights-of-ways for actions like transmission lines or pipelines], Livestock grazing and rangeland management [two BLM categories that we combined into one for our analysis because of significant overlap in the decisions in the two categories], Mining, and Recreation and visitor services [hereafter, Recreation]) and 10 EAs from all other proposed action categories with completed EAs in Colorado during this time period combined (Conservation and preservation areas, Other, Paleontology, Vegetation, Wild horses and burros, and Wildland fire management) for a total sample of 70 EAs (16% of the 426 BLM Colorado EAs total that met our inclusion criteria, see [27] for a full list of the 70 EAs sampled).

Within each sampled EA, we then recorded the types of resources analyzed, using a list of 48 resource categories developed by the BLM. EAs generally analyze the potential effects of a proposed action on the environment on a resource-by-resource basis. We considered resources to have been analyzed if they had a section header in the main body of the document and at least one sentence analyzing the resource (e.g., describing how the agency determined whether the resource was present in the area and, if present, how and in what ways the resource might be affected by the proposed action).

Next, we identified and counted all citations in the following sections of the main body of the EA: Affected Environment, Environmental Effects/Consequences, Cumulative Effects, and Past, Present, and Reasonably Foreseeable Future Actions. We did not analyze or record citations from other sections of the EA that are not related to the environmental effects analysis (e.g., Introduction, Purpose and Need, Scoping, Proposed Action and Alternatives) or from appendices. To be considered a citation for the purposes of this study, the citation needed to be written in a standard format that included the author or document title and year, and to clearly reference a specific document in the References Cited section.

We then categorized all citations meeting our criteria for inclusion by age and document type. The age of the citation was determined based on the year that the document was published or finalized. We categorized age as follows: 2015–2019, 2010–2014, 2000–2009, 1980–1999, and 1979 and earlier. The 2015–2019 and 2010–2014 age categories were combined into a single category (2010–2019) for all figures. We used a decision tree (S1 File) to categorize each citation as one of 16 types of documents (Table 1) based on the type of information (data, science, policy or management, or other), whether the document was authored by the BLM or by another entity, and whether the document was published. For this study we were primarily interested in citations of data and science products, but we wanted to be comprehensive in our consideration of citations. USGS and BLM developed the decision tree together, using

**Table 1. The sixteen types of documents into which we categorized citations.** These document types were used to assess the number, type, and age of documents cited in 70 Environmental Assessments completed by the Bureau of Land Management (BLM) in Colorado from 2015–2019.

| Document type | Broad category of document | Description of document type | Examples |
|---|---|---|---|
| Published BLM inventory or dataset | Data | Published or publicly-available data or inventories, including maps and descriptions if a non-quantitative description is appropriate (e.g., cultural resource surveys, visual resource descriptions) that is collected by the BLM. | BLM Assessment, Inventory, and Monitoring (AIM) data, BLM grazing allotments |
| Unpublished BLM inventory or dataset | Data | Unpublished data or inventories, including maps and descriptions if a non-quantitative description is appropriate (e.g., cultural resource surveys, visual resource descriptions) that is collected by the BLM. | BLM cultural resource inventories, BLM rare plant surveys |
| Published non-BLM inventory or dataset | Data | Published or publicly-available data or inventories, including maps and descriptions if a non-quantitative description is appropriate (e.g., cultural resource surveys, visual resource descriptions) not collected by the BLM. | National Resource Conservation Service (NRCS) soils data, LANDFIRE data, National Land Cover Database (NLCD) data |
| Unpublished non-BLM inventory or dataset | Data | Unpublished data or inventories, including maps and descriptions if a non-quantitative description is appropriate (e.g., cultural resource surveys, visual resource descriptions) not collected by the BLM. | Cultural resource inventories completed by contractors |
| Journal article | Science | Documents from legitimate, non-predatory journals. The majority of these were peer-reviewed, although we did not specifically search to determine if journals involved peer-review. | Articles in open-access or peer-reviewed journals |
| BLM science report | Science | Documents that report original science findings or summaries or syntheses of science findings research that are peer-reviewed or have a Digital Object Identifier (DOI) and are authored by the BLM. | BLM Technical Reports |
| Non-BLM science report | Science | Documents that report original science findings or summaries or syntheses of science findings research that are peer-reviewed or have a DOI and are not authored by the BLM. | U.S. Geological Survey Open-File Reports, U.S. Forest Service General Technical Reports, U.S. Geological Survey Scientific Investigator Reports, U.S. Geological Survey Bulletins, U.S. Forest Service science reports, university white papers |
| Other publicly-available BLM report | Science | Reports where it is not clear that the report underwent a peer-review process that are authored by the BLM. | BLM Allotment Master Reports, other BLM reports |
| Other publicly-available non-BLM report | Science | Reports where it is not clear that the report underwent a peer-review process that are not authored by the BLM. | Reports completed by contractors |
| Other science product | Science | Non-peer-reviewed science information including books, conference proceedings, theses, dissertations, and newspapers. | Book chapters, conference papers, graduate theses |
| Law, policy, manual, or guidance | Policy or management | Any document that provides guidance on how to do something. Many of these documents provide guidance to the agency or include statements of rules that BLM must follow. | NEPA handbook, BLM Instructional Memos, BLM Information Bulletins, Secretarial Orders, Executive Orders |
| Science monitoring plan or strategy | Policy or management | These documents include National level strategies, plans for national monuments or National Conservation Areas, recovery plans for specific threatened and endangered species, or descriptions of BLM data and research programs. | The Integrated Rangeland Fire Management Strategy Actionable Science Plan, National Seed Strategy, The Greater Sage-Grouse Monitoring Framework, BLM Assessment, Inventory, and Monitoring Strategy |
| BLM plan to manage a resource | Policy or management | Decision documents completed through the NEPA process, as a part of land-use planning, documents to guide management of a resource or program area, or appendices or addendums to any of these documents completed by the BLM. | Environmental Impact Statements and other NEPA documents, Resource Management Plans, Biological Assessments, Reasonably Foreseeable Development Scenarios, or species conservation strategies completed by the BLM |
| Non-BLM plan to manage a resource | Policy or management | Decision documents completed through the NEPA process, as a part of land-use planning, to guide management of a resource or program area, or appendices or addendums to any of these documents completed by any agency or organization outside of the BLM (e.g., U.S. Fish and Wildlife Service, U.S. Forest Service). | Environmental Impact Statements and other NEPA documents, Resource Management Plans, Biological Opinions, Reasonably Foreseeable Development Scenarios, or species conservation strategies completed by another entity (e.g., U.S. Fish and Wildlife Service, state agency) |
| Website | Other | A website that does not meet the criteria for any of the other document types. | County website describing local economy |
| Unclear | Other | Any document that cannot be coded into one of the proceeding categories. | |

non-overlapping categories that would be informative to the agency and specific to the types of science information that BLM staff typically use in their NEPA analyses. For citations that we could not categorize based on the information in the References Cited section of the EA, we spent up to five minutes searching for the document using the Google search engine to obtain more information. One researcher coded the EAs after a period of practice sessions and consultation with the author group to establish clear understanding, repeatability, and coding consistency. When questions arose about coding specific citations, we use negotiated agreement to decide collectively how to code the citation [38].

We summarized our findings by both the type of proposed action and by resource, as BLM decisions, programs, and funding are organized and categorized by both. This level of detail was necessary for our results to be meaningful for BLM program leads and resource managers, who may be focused on specific actions or resources. We performed analysis of variance and covariance tests in R (version 4.2.2) using the packages 'dplyr' and 'car' using a Type III sum of squares because of the unbalanced nature of the data. We tested 1) if the total number of citations per EA differed by action and by BLM office (and the interaction between the two factors) when considering differences in the number of resources analyzed in each EA, and 2) if the number of citations per resource analysis section differed by resource and by BLM office (and the interaction between the two factors). We included in our statistical tests and present below action-specific results only for the six categories of actions that were analyzed 10 times in our sample, and resource-specific results only for those resources that were analyzed at least five times in our sample. We sometimes refer to these as commonly proposed actions and commonly analyzed resources to ensure clarity. We include resource-specific results about the age and type of citations only if there were at least 10 citations in our sample for that resource.

For the statistical test examining the relationship between action and BLM office on the total number of citations per EA, we excluded three EAs from the analysis that were completed in BLM offices that only completed a single EA for a commonly proposed action. We retained all other EAs (n = 58), which were all completed by a BLM office (n = 8) that completed at least two EAs (range 2–17 EAs per BLM office) across a minimum of two commonly proposed action categories (range 2–5). For the statistical test examining the relationship between resource and BLM office on the total number of citations per resource analysis section, our dataset included 13 BLM offices, each of which had completed a minimum of four resource analysis sections (range 4–243 resource analyses per BLM office) across a minimum of four different types of resources (range 4–33), resulting in a sample size of 688 resource analysis sections across 33 types of resources. The interaction between action and BLM office was not significant (p = 0.89), so it was removed from the final model. We used a significance level of α = 0.05 for all statistical tests.

## Results

### Overall characteristics of citations

We found a total of 1,187 citations that met our criteria for inclusion within our sample of 70 BLM EAs in Colorado from 2015–2019. The mean number of citations per EA was 17 (range 0–111). Most citations were recent: 28% were from the five years preceding our study period (2015–2019), nearly half (47%) were from the last decade (2010–2019), and 78% were from 2000 or later (Fig 1). Most citations were categorized as science (50%), with fewer categorized as data (23%), policy or management (22%), and other (4%). On average, four data documents and eight science documents were cited in each EA. Journal articles were the most frequently cited type of document (26% of citations), followed by published, non-BLM inventories or

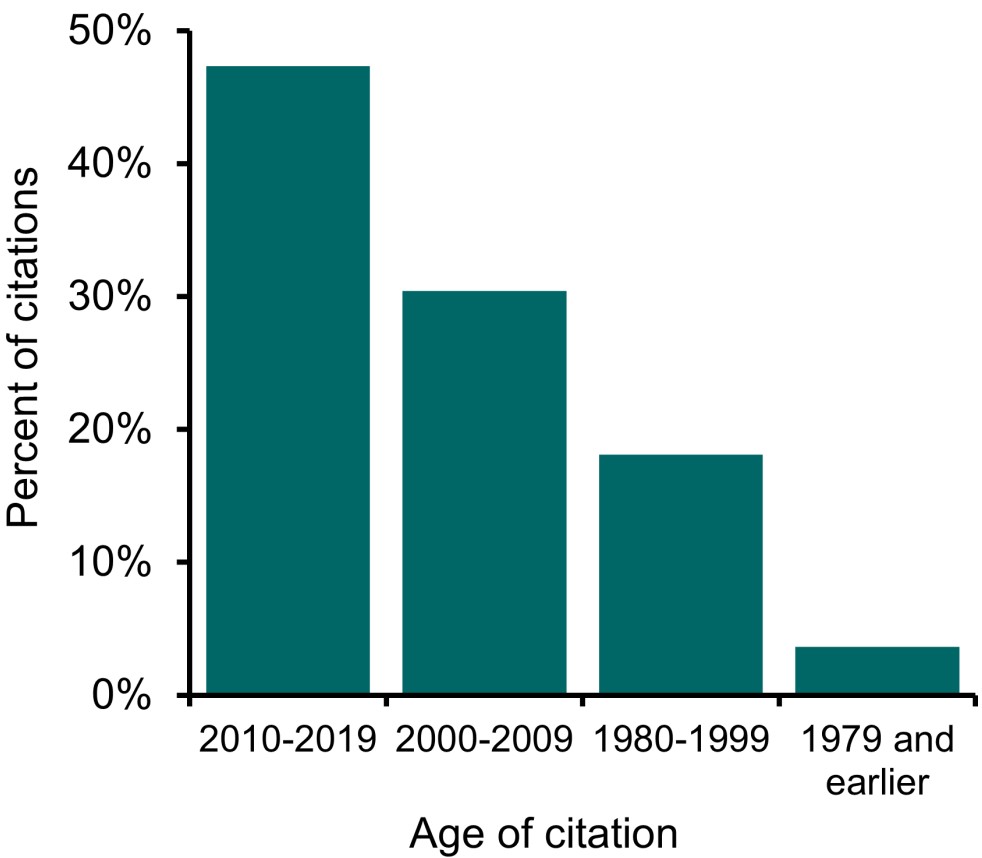

**Fig 1. Age of cited documents in Bureau of Land Management (BLM) Environmental Assessments.** The documents presented are from a stratified random sample of 70 Environmental Assessments completed by the BLM in Colorado from 2015–2019. Six citations of unknown age are not included in the figure.

datasets (13% of citations), and law, policy, manual or guidance documents (12% of citations, Fig 2). The number of citations per EA differed significantly by both action (p = 0.01) and BLM office (p = 0.02), while accounting for a significant covariate: the number of resources analyzed in each EA (p < 0.001).

## Characterizing citations by proposed action

The commonly proposed actions with the most citations per EA were Oil and gas development (mean of 41 citations per EA, range 0–90), Fish and wildlife (mean 33, range 0–111), and Lands and realty (mean 13, range 0–59; Fig 3). EAs analyzing Recreation actions had the fewest citations per EA (mean 6, range 0–18), followed by Livestock grazing and range management (mean 8, range 0–35).

With regard to age of citations, Mining actions had the highest proportion of citations from the most recent time period (38% of citations were from 2015–2019), followed closely by Lands and realty (35% of citations from 2015–2019). Recreation actions cited the greatest proportion of older documents, with 27% of citations dating from 1999 or earlier.

Generally, EAs cited science documents more frequently than data documents. For example, 73% of citations in Recreation EAs were science documents. Data citations comprised 16–30% of all citations in EAs for each of the commonly proposed types of actions.

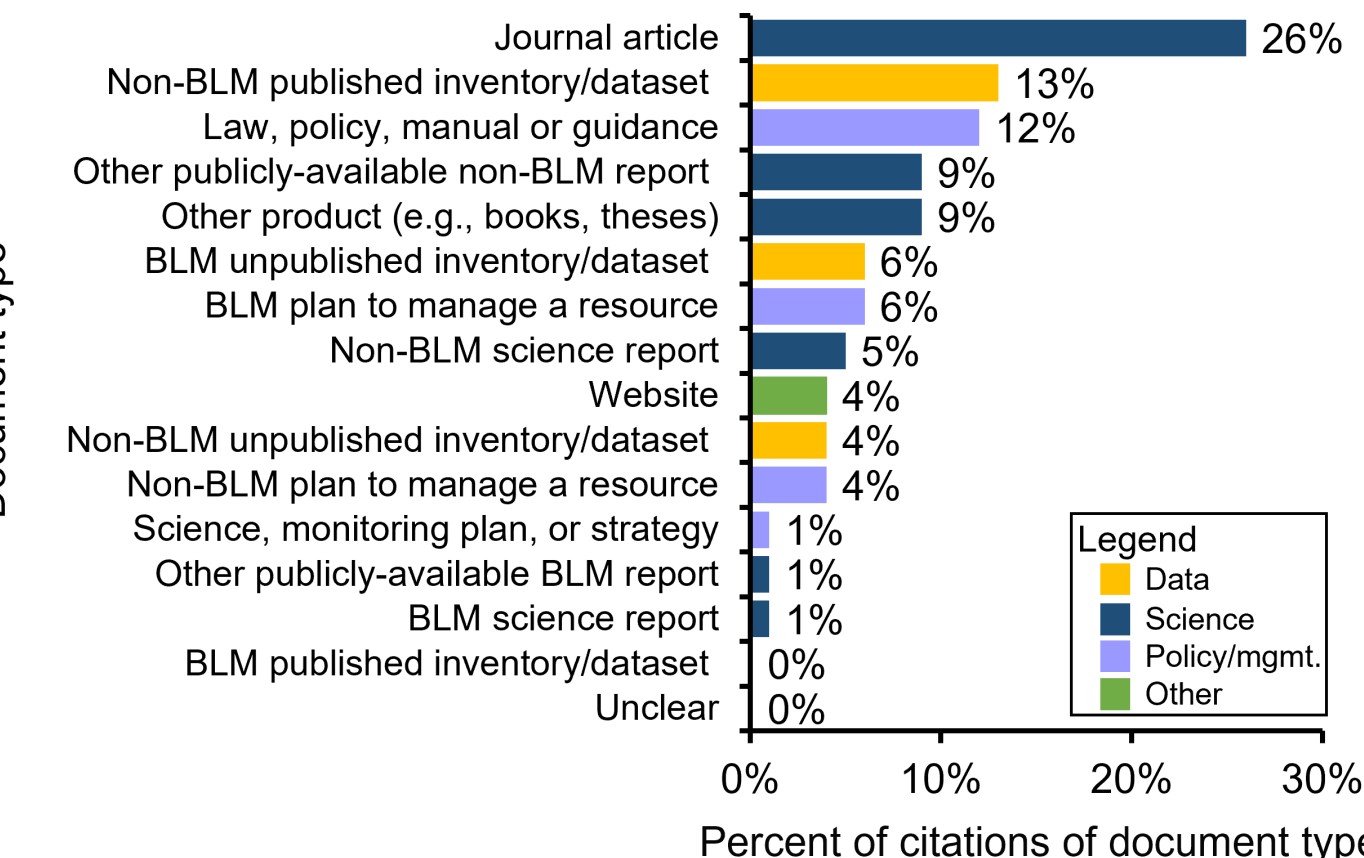

**Fig 2. Types of science and data documents cited in Bureau of Land Management (BLM) Environmental Assessments.** The documents presented are from a stratified random sample of 70 Environmental Assessments completed by the BLM in Colorado from 2015–2019. See Table 1 for a description of each document type.

The specific types of documents cited differed by action. Oil and gas development EAs had the highest citation rate of journal articles (mean of 11 citations per EA). Comparatively, Livestock grazing and range management documents cited on average one journal article per EA. Unpublished, BLM inventories or datasets (21% of citations) were the most common type of citation in Livestock grazing and range management EAs.

### Characterizing citations by affected resource

Forty-five percent of individual resource analysis sections (315 of 702 sections total) contained at least one citation (of any type). The mean number of citations per resource analysis section differed by resource ($p = 0.03$), with the effect of resource varying by BLM office (i.e., a significant interaction between resource and BLM office, $p < 0.05$). Twenty-one percent of resource sections (149 of 702) contained at least one data citation, and 24% (165 of 702) contained at least one science citation.

Of the 1,187 total citations coded, the following commonly analyzed resources (i.e., resources analyzed at least five times in our sample) had the highest mean number of citations: socioeconomics (mean of 5.1 citations per analysis section, range 0–12), mineral resources (mean 4.5, range 0–15), noise (mean 4.0, range 2–5), sage-grouse (*Centrocercus urophasianus* and *Centrocercus minimus*; mean 3.7, range 0–21), geology (mean 3.2, range

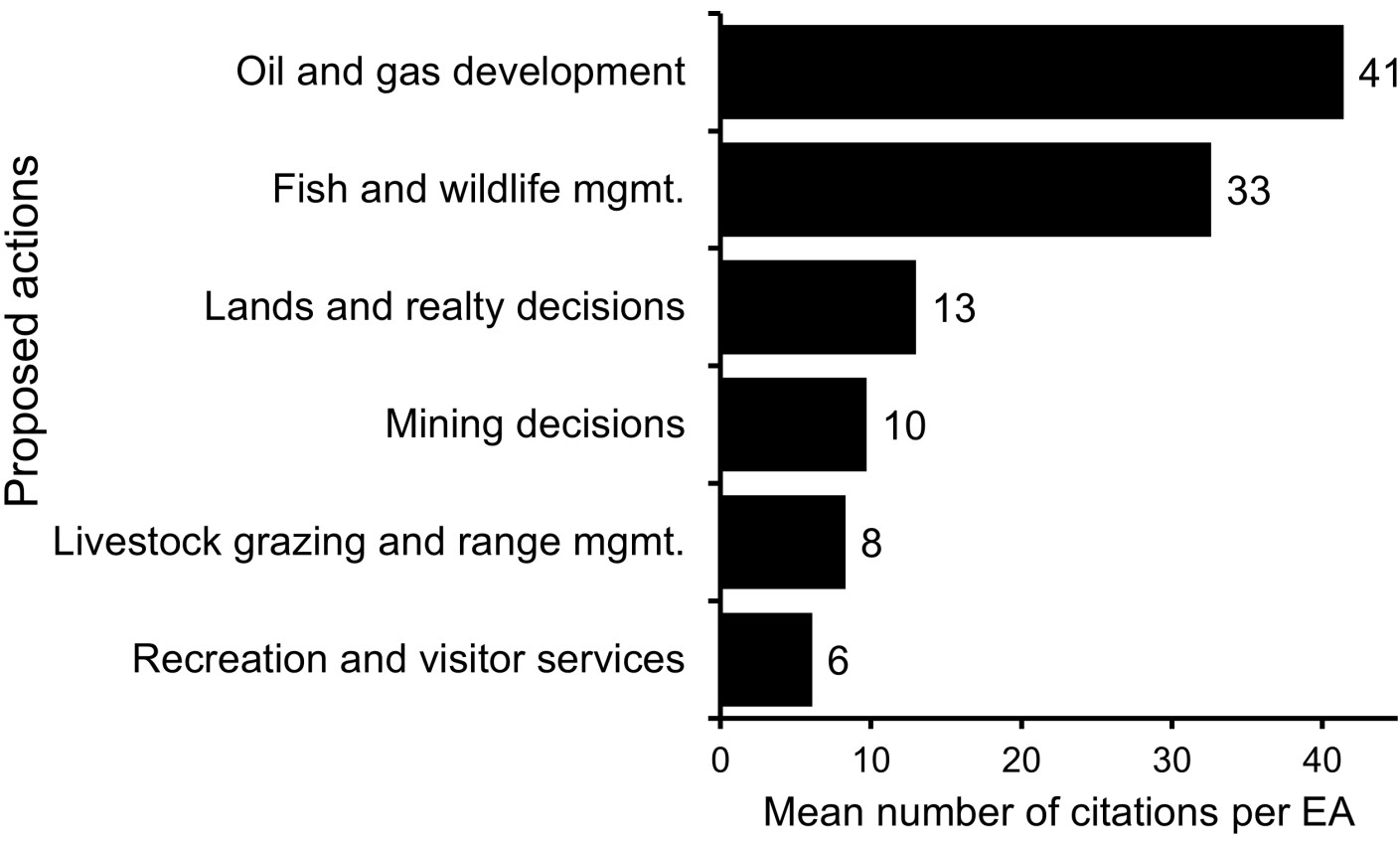

**Fig 3. Mean number of citations per Bureau of Land Management (BLM) Environmental Assessment by category of proposed action.** The documents presented are from a stratified random sample of 70 Environmental Assessments (EAs) completed by the BLM in Colorado from 2015–2019. Proposed actions that were analyzed in fewer than 10 EAs (Wildland fire management, Wild horses and burros, Conservation and preservation areas, Paleontology) are not shown in the figure.

0–9), and 'air quality and climate' (mean 3.1, range 0–16; Fig 4). Fourteen of 33 commonly analyzed resources included less than one citation on average (Fig 4), including native peoples' resources (items or land that Native Americans identify as resources of importance), which were analyzed in 34% of all EAs but did not contain a citation in any analysis section.

In terms of age of citations, sections analyzing potential effects of a proposed action on existing recreation had the highest percentage of recent citations (70% of citations were from 2015–2019; S11 Fig). On the other hand, sections analyzing potential effects to geology most frequently cited documents from 1979 or earlier (50%; S5 Fig), with the oldest citation dating back to 1927.

Only three of 33 resources that were commonly analyzed included at least one data citation on average (socioeconomics, 'air quality and climate', and soils; S13, S1, and S14 Figs). Commonly analyzed resources with no data citations were 'fire ecology and management' and 'grazing and range' (S4 and S6 Figs). Soils had the greatest proportion of citations that were categorized as data (71%; S14 Fig).

Twelve of 33 resources that were commonly analyzed included at least one science citation on average. Vegetation and 'wild horses and burros' cited science documents most frequently (85% of all citations for both resources were science documents), followed by mineral resources (80%; S19, S23, and S8 Figs). Vegetation had the highest proportion of journal

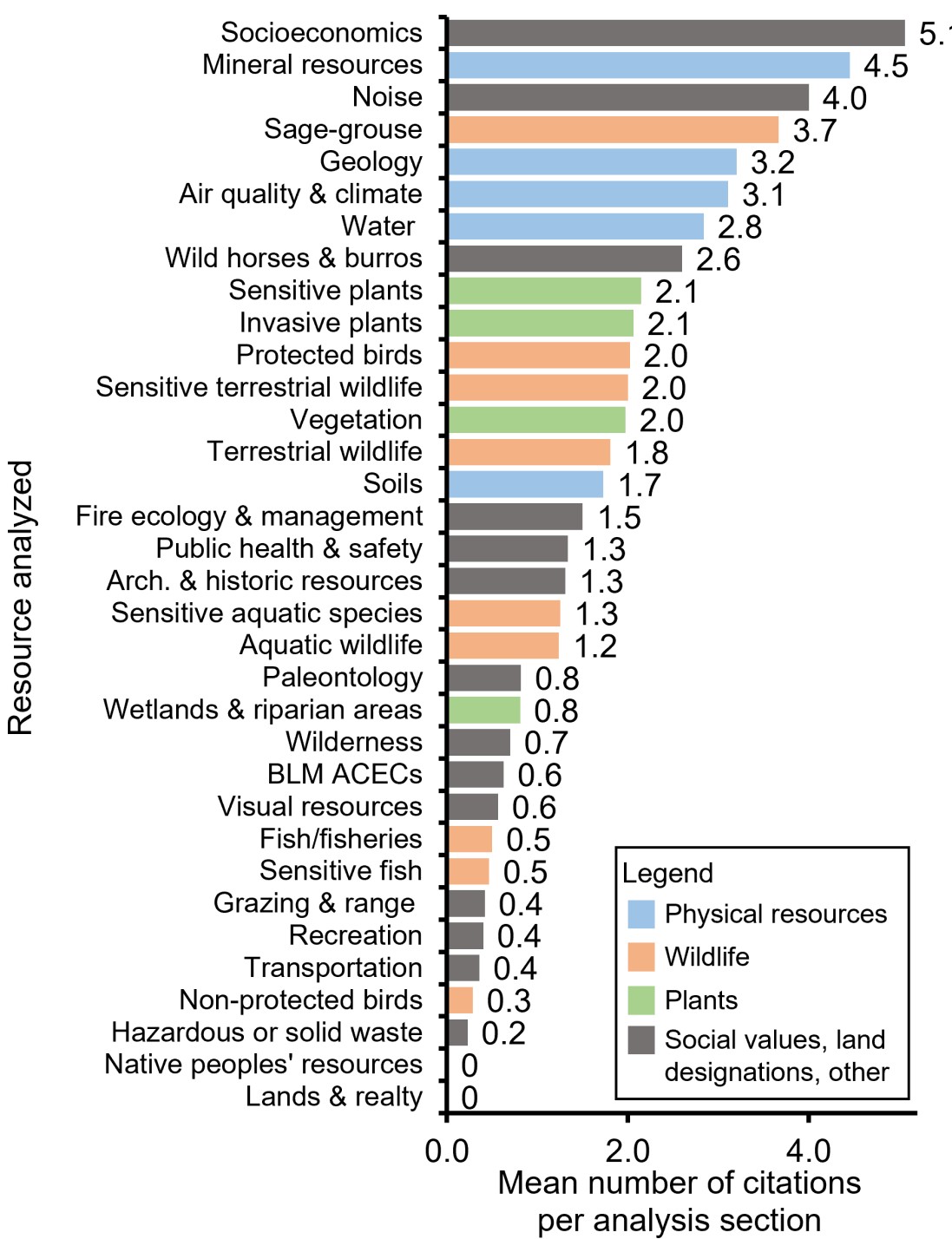

**Fig 4. Mean number of citations per resource analysis section in Bureau of Land Management (BLM) Environmental Assessments.** The documents presented are from a stratified random sample of 70 Environmental Assessments completed by the BLM in Colorado from 2015–2019. Resources shown are those that were analyzed in five or more Environmental Assessments. Colors correspond to general classes of resources: blue, physical resources (e.g., water, minerals, air); orange, wildlife; green, plants; and gray, social values, land designations, and other resources (e.g., noise, wilderness, archaeological and historic resources). Abbreviation is as follows: ACEC, Area of Critical Environmental Concern, which is a land designation in the BLM for areas where special management is needed (e.g., unique scenic landscapes, important resources).

article citations (71%), followed by 'wild horses and burros' (69%), and invasive plants (64%; S19, S23, and S7 Figs).

## Discussion

The use of science and data in decisions involving public lands and resources is often required by law (e.g., NEPA, Endangered Species Act), and challenges to these decisions include challenges to the agency's use of science and data in the decision process [14]. Citing science and data sources is the most transparent way for agencies to demonstrate their use of science and data in decision-making. Citing published and publicly available data and documents can increase the transparency and defensibility of decisions and build stakeholder trust in the decision process [39]. Citations of recent published work about potentially affected resources can also help to demonstrate use of credible, reliable science, which is required by NEPA (40 CFR §§ 1502.21(d), 1502.23), though it is important to note that older science may still be relevant and of high quality, and more recent science is not valid simply by virtue of recency (it must still be robust). We identified the number, type, and age of documents cited within a sample of 70 recent EAs completed by the BLM in Colorado. We found that EAs included an average of 17 citations, but that the number of citations was highly variable (range 0–111). Most citations were categorized as science or data (73%), and about half were recent (47% were from 2010 or later). EAs related to Oil and gas development actions and analyses of socioeconomic effects most frequently included data and science citations in our sample.

### Characterizing citations in BLM Colorado EAs

While the concept of assessing the evidence base for natural resource decisions and plans is not new, few studies look in-depth at documents cited. The only study to our knowledge that has assessed the type of cited documents in Federal land management decisions categorized scientific citations in Forest Plans completed by four U.S. Forest Service National Forests in Alaska, New Mexico, California/Nevada, and North Carolina [32]. They found journal articles to be frequently cited (23% of all citations in their sample), along with technical reports (39% of all citations). In our study, journal articles were the most frequently cited document type (26% of citations), and 6% of citations were technical reports (though our definition of a technical report was narrower than that used by [32]). Frequent citation of journal articles and technical reports (government technical reports are typically peer-reviewed and published) likely represents an effort by land managers to use the best available science in their plans and decisions. [40] found that Federal land managers considered peer-reviewed scientific publications to be the "best available science."

We are not aware of other studies that have explored the age of documents cited in the environmental analyses that inform public lands decisions. We contend that age is important to understanding whether the best available science and data are being used to inform decision-making, as the scientific process addresses limitations, uncertainties, and inconsistencies over time [20]. We found that while most citations were relatively recent, 22% of citations were to documents more than 15 years old (i.e., prior to 2000). We suggest that scientists particularly consider topics for which science is evolving rapidly (e.g., effects of energy development on wildlife) and develop science summary and science synthesis products that can facilitate incorporation of more recent findings into public lands decision-making.

### Characterizing citations by proposed action

Across our entire sample, BLM EAs in Colorado included an average of 17 citations, with the mean number of citations varying by action, BLM office, and a covariate - the number of

resources analyzed in the EA. Oil and gas development EAs had the greatest mean number of citations per document. This may reflect a focus on science use and defensibility in Oil and gas development decisions: decisions on Federal, multiple-use public lands are frequently litigated [41,42], and BLM in this region is litigated on its use of science and data most frequently for Oil and gas decisions [14]. Livestock grazing decisions are also frequently challenged on the basis of science or data [14], but had substantially fewer citations per EA.

EAs for each of the commonly analyzed proposed action categories in our sample included an average of at least six citations. However, all commonly analyzed categories of proposed actions also had at least one EA (and up to five of the ten sampled) that contained no citations. In the BLM, categories of actions typically correspond to agency programs, such as the recreation, livestock grazing, and fluid minerals (oil and gas) programs, which are administered at a national level. We also found that the number of citations varied by BLM office, suggesting that science and data use in decision-making, and documentation of that use in EAs, also varies by administrative units (e.g., BLM field offices) within programs. Citations might also vary by the individual developing the EA, as multiple individuals typically contribute to any single EA, and in some cases, some analyses may even be completed by external contractors. Other studies have suggested that use of information by resource managers is largely driven by their experience and personal connections [33,40].

## Characterizing citations by affected resource

The mean number of citations for individual resource analysis sections varied significantly by resource and by BLM office. Sections analyzing potential effects to socioeconomics had the greatest number of citations on average (5.1 citations per resource analysis). Interestingly, this conflicts with a study that surveyed BLM and U.S. Forest Service managers about challenges in ecosystem management and found that integration of socioeconomics information into decision-making was perceived to be difficult [43]. Integration of socioeconomic information into decisions is a current priority for the agency [44,45] including for the purpose of meeting new federal environmental justice goals (e.g., [46]).

Interestingly, more frequent use of citations did not mean that a higher number of unique documents were cited. In the case of noise analyses, for example, while we counted a total of 28 citations for the seven noise sections that we analyzed, only eight unique documents were cited in these analyses (they included three document types: data, science, and policy or management documents). Other studies have also found that the same analyses and supporting documents may be used in multiple management decisions [30,47]. Citing a relatively small number of core documents across multiple NEPA analyses can happen for multiple reasons, including that the proposed actions, and thus the relevant science, may be similar. The same individual or group may also conduct the NEPA analysis for multiple projects. Both scenarios highlight the importance of scientists and management agencies working together to ensure that the science and data most accessible to and used by resource managers reflects the best available information on that topic.

For most resources, we might typically expect to see at least two citations in the NEPA analysis section for that resource: one data citation, reflecting what is known about the presence or condition of the resource at the site, and one science citation, reflecting what is known about how that resource may be affected by the proposed action [27]. While 13 of 33 (39%) commonly analyzed resources contained at least two citations on average in the analysis section for that resource, more than half of the commonly analyzed resources (20 of 33) in these EAs had fewer than two citations. Some of these resources may be more likely to rely on other types of knowledge, such as traditional ecological knowledge, which would be less likely to be

formally cited. For example, we recorded no citations for native peoples' resources, though 34% of the EAs in our sample analyzed that resource. [33] also found that information about the condition of indigenous cultural heritage was primarily based on resource managers' experience. For other resources, agency staff may be more likely to rely on field surveys or professional expertise to inform the analysis, and thus there would not be an associated formal citation.

There are also multiple reasons that resource managers may not have cited documents in their analysis of a particular resource even if those documents exist, including that they used science and data but did not specifically cite those datasets or documents in the EA. A lack of access to relevant data or science [11] or institutional or policy limitations (e.g., imposed time and page limits on NEPA analyses that might discourage staff from taking the time and document space to include formal citations [40]) may also lead to few science and data citations in NEPA analyses. [48] found that primary barriers to the integration of climate change science into Federal decision-making included changing agency guidance along with time and resource limitations. Researchers with the U.S. Forest Service have documented a purposeful lack of citations in decision documents in an attempt to limit potential challenges, reporting that managers believe that they will never have all the data that they need and that citing data may make their decisions more vulnerable to public comment or challenges [49]. Federal land managers participating in two studies [30,34] also stated that the scientific literature was "too time consuming to read," "too technical or difficult to interpret in the context of their decision-making," or that "they rely on 'in-house' advisors or expert groups to interpret information". Additionally, the increasing number of papers published each year can make it difficult to keep up with the current scientific literature [50].

Our study had limitations, including that we studied a sample of one specific type of decision analysis (EAs) made by a single agency (the BLM) in a single state (Colorado). However, our findings do reflect decisions about multiple types of actions and their potential effects to many resources that occur on public lands across the United States [27]. We also note that we analyzed only publicly available documents: EAs posted on BLM's National NEPA Register. Additional science information and citations may be contained in appendices to these EAs and in the full administrative record associated with the decision (administrative records are not publicly available). We did not remove duplicate citations in this study. Multiple citations of a single document may refer to separate findings within that document, especially in the case of citations of literature reviews or long reports. Thus, we counted each citation regardless of whether that document had been cited before in that EA. Future studies might delve further into the nature of cited science documents (e.g., primary literature, summary products such as annotated bibliographies, review or synthesis products, metanalyses) and into how the number and type of science and data citations included for different topics may relate to the available evidence base or level of contention associated with different types of actions and resources.

Our findings provide a foundational step in understanding current use of science and data in public lands decision-making. They also provide insight into the extent to which cited documents may reflect transparent use of the best available science and data about specific actions and resources. Scientists can use these findings to help identify specific resources and actions for which use of recent, published science and data in public lands decision-making could be strengthened and made more transparent.

Science providers are working to develop products to help public land managers more easily access, use, and cite science and data in their decision documents and analyses. The U.S. Geological Survey is compiling, summarizing, and synthesizing scientific information on topics that are a high priority for public land managers (e.g., Greater sage-grouse

(*Centrocercus urophasianus*, [51,52]), invasive annual grasses [53], and the effects of energy development on wildlife [54]) and working to share that information in a searchable format that also allows for easy download and incorporation by reference into agency NEPA analyses and decision documents (https://apps.usgs.gov/science-for-resource-managers/#/). Researchers are also compiling and comparing available datasets on invasive annual grasses [55], and evaluating remotely sensed data products to promote their appropriate use in public lands decisions [56]. Similar efforts have been initiated by other agencies and organizations (e.g., fire science, [57]). Such efforts that focus on compiling and synthesizing science for public land managers on the specific topics that they analyze most frequently [27] is one avenue for helping to increase the transparency, defensibility, and science foundation of public lands decision-making.

## Supporting information

**S1 File. Decision tree that was used to categorize the types of documents cited within BLM Environmental Assessments.**
(PDF)

**S2 File. Supporting figures.** This file contains figures of the age of citation and type of documents cited for all resources that 1) were analyzed five or more times across all Environmental Assessments and 2) had a total of at least ten citations across all analysis sections for that resource: Air quality and climate (S1 Fig); Aquatic wildlife (S2 Fig); Archaeological and historic resources (S3 Fig); Fire ecology and management (S4 Fig); Geology (S5 Fig); Grazing and range (S6 Fig); Invasive plants (S7 Fig); Mineral resources (S8 Fig); Noise (S9 Fig); Protected birds (S10 Fig); Recreation (S11 Fig); Sage-grouse (S12 Fig); Socioeconomics (S13 Fig); Soils (S14 Fig); Sensitive aquatic wildlife (S15 Fig); Sensitive plants (S16 Fig); Sensitive terrestrial wildlife (S17 Fig); Terrestrial wildlife (S18 Fig); Vegetation (S19 Fig); Visual resources (S20 Fig); Water (S21 Fig); Wetlands and riparian areas (S22 Fig); Wild horses and burros (S23 Fig).
(PDF)

## Acknowledgments

We are grateful to Tye Morgan, Karen Prentice, Megan McLachlan, Steve Hanser, David Wood, Matt Preston, Russell Slatton, and to multiple staff at the BLM Colorado State Office, including Chris Domschke, Sam Dearstyne, Leigh Espy, Jennifer Montoya, and Brittany Sprout, for input on methods development and insights on interpreting results for this project. Jennifer Meineke, Ella Samuel, Mona Khalil, Benjamin Gaddis, and Madeline Scheintaub provided helpful reviews of earlier drafts of this manuscript. We are grateful to Andrew Fayram for statistical assistance. Any use of trade, firm, or product names is for descriptive purposes only and does not imply endorsement by the U.S. Government.

## Author contributions

**Conceptualization:** Alison C. Foster, Sarah K. Carter, Travis S. Haby.

**Data curation:** Alison C. Foster, Andrew T. Canchola, Sarah K. Carter.

**Formal analysis:** Alison C. Foster, Andrew T. Canchola, Sarah K. Carter.

**Funding acquisition:** Sarah K. Carter.

**Methodology:** Alison C. Foster, Andrew T. Canchola, Sarah K. Carter, Travis S. Haby.

**Project administration:** Alison C. Foster, Sarah K. Carter.

**Supervision:** Alison C. Foster, Sarah K. Carter.

**Visualization:** Alison C. Foster, Andrew T. Canchola.

**Writing – original draft:** Alison C. Foster, Andrew T. Canchola, Sarah K. Carter.

**Writing – review & editing:** Alison C. Foster, Andrew T. Canchola, Sarah K. Carter, Travis S. Haby.

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
