## [Decision Letter · Decision Letter 0]

10 Jul 2024

PONE-D-24-09193Exploring the science and data foundation for Federal public lands decisionsPLOS ONE

Dear Dr. Foster,

Thank you for submitting your manuscript to PLOS ONE. After careful consideration, we feel that it has merit but does not fully meet PLOS ONE’s publication criteria as it currently stands. Therefore, we invite you to submit a revised version of the manuscript that addresses the points raised during the review process.

I've received sets of comments from two reviewers. Based on their assessments as well as my own reading, I think that your manuscript could be suitable for publication with some relatively modest changes. Foremost, I would like you to consider and address the points raised by Reviewer 1, which -- as was their stated intention -- should further strengthen an already strong manuscript. Overall, your manuscript is well written, so the revisions will mostly involve minor clarifications or additions to your existing material.

We look forward to receiving your revised manuscript.

Kind regards,

Frank H. Koch, PhD

Academic Editor

PLOS ONE

Journal Requirements:

**Additional Editor Comments:**

Specific comments:

Line 77 - delete comma after "Independent" (I don't think that it's necessary)

Table 1 - in the first two "Description of document type" boxes, change "that is" to "that are"

Line 402 - "socioeconomic" instead of "socioeconomics"

In the supplemental information, I suggest adding a title for each S1-S23 Fig., so readers don't have to flip back to the master list on p.5.

Reviewers' comments:

Reviewer's Responses to Questions

**Comments to the Author**

1. Is the manuscript technically sound, and do the data support the conclusions?

Reviewer #1: Yes

Reviewer #2: Yes

2. Has the statistical analysis been performed appropriately and rigorously? 

Reviewer #1: I Don't Know

Reviewer #2: Yes

3. Have the authors made all data underlying the findings in their manuscript fully available?

Reviewer #1: Yes

Reviewer #2: Yes

4. Is the manuscript presented in an intelligible fashion and written in standard English?

Reviewer #1: Yes

Reviewer #2: Yes

5. Review Comments to the Author

Reviewer #1: Thank you for the opportunity to review this manuscript. It is a well-written and clear contribution the emerging literature on the use of science in agency decision-making. The authors gather and code citations from 70 Environmental Assessments (EAs) from the BLM in Colorado from 2015-2019. They analyze the age, type, and frequency of citations across (a) proposed actions and (b) resources that could be affected by the proposed actions. The manuscript provides a rich scholarly lens in which to contextualize new descriptive research findings. Overall, the manuscript is strong. However, I have a few comments and questions that I hope help to improve the contribution:

1) Given that the “introduction” section covers a lot of theoretical material and prior research (and is less of an outline), I suggest titling it according to the content or puzzle the authors address in the manuscript. Alternatively, they could perhaps include a very brief introduction section and save the bulk of the literature for a section on that topic.

2) Potentially one of the more important comments among my comments/questions: In the introductory paragraph starting on line 71, the authors define what they (and the law) mean by “best available science”. While they acknowledge that “different groups can have different perceptions of what is most important in characterizing best available science”, the authors do not dive any deeper into the literature on the bias toward Western science and ways of knowing. Given the recent discussions by federal agencies, universities, and researchers on the bias toward Western knowledge production and use in policy and management, the authors should spend a paragraph or two addressing this issue. This seems especially important given the authors’ later focus on the lack of Indigenous perspectives.

3) The authors justify the study case, in part because of their established relationships with BLM staff. Can the authors address what may have been lost had the study not engaged a co-production model? The approach certainly seems to provide benefits in terms of the ability to build capital and share findings with decision-makers, but the authors should address what was gained through partnership (perhaps things like better understanding citation types).

4) The authors focus on EAs. Besides EAs being more common, why not EISs as well? Do the authors expect any differences between the two types of analyses?

5) The authors focus on citations within only the “main body” of the EA, excluding information from the “purpose and need” and other sections as well as from Appendices.

a. First, I think it would be very interesting to know what citations come up in the “purpose and need” section because this section frames the problem – and could therefore be particularly important. However, I understand this may be outside the scope of the current analysis.

b. Second, my hunch is that a lot of citations are included in the Appendices. The authors could take a sub-sample of their documents and examine the extent that Appendices include citations to address this issue. If not that, are there reasons to be concerned that the science is more recent, a different type, etc. in the Appendices? Or that citations look different across resources or action purpose in Appendices?

6) The authors analyze 10 EAs from several proposed action categories. What is the percentage of total EAs by proposed actions? I ask because I’m tryng to get a better sense of generalizability.

7) The table laying out the types of citations is both impressive and a contribution! As is the decision tree/codebook in the supplemental material. Do the authors envision this codebook to be used to analyze citations in the BLM more broadly or other agencies? I suggest the authors talk their typology up a bit more in the conclusion.

8) Was intercoder reliability checked to ensure coding concepts were clear and consistent?

9) I get a little lost, in multiple places, between the measures on proposed actions versus resources. The section titles are similar, the figures in the supplementary materials only address resources, and so on. I do not have a strong suggestion for improvement (and apologize for offering critique without assist), but think the authors should spend more time thinking through how to discuss and visualize their analyses across these two dimensions.

a. As an example, the authors write that recreation had the oldest literature (roughly line 281) in one section and the newest literature (roughly line 381) in another section.

b. Along these lines, the authors may consider some re-organization, for example presenting results and discussion on proposed actions and then on resouces, if those distinctions remain the most important.

c. Relatedly, I am a bit confused on the use of and outcomes of the statistical tests (page 13 especially). There seems to be a lot of exclusions applied. (And statements that seem at odds with one another, such as the note that office was not significant and thus removed and in the results section that it was significant). On the whole, I wonder how worthwhile the use of statistical tests is in a largely descriptive analysis. If the authors feel the need to test for difference, which I think is fine if the n is large enough, then the next version of the paper should make the approach and outcomes clearer. Perhaps a visual demonstrating the statistical comparisons and data subsetting would help?

10) I’m not sure all the suppl. figures on resources are needed in a peer-review publication, although I respect that they are needed by BLM partners.

11) In line 348, the authors write that most citations were categorized as “science” but I’ve forgotten at this point what “science” means in terms of the categories.

12) I was surprised to read near the end that unique references weren’t being counted. I suggest the authors focus on unique references more strongly, perhaps in comparison to citations (repeated references).

13) In lines 411-414, the authors discuss explanations for a small number of core citations. They may also want to consider the likelihood that EA text and references are being recycled. See for example Hileman et al. (2021).

14) The authors have some particularly important findings on the representation Indigenous values and interests in the EAs. I think the authors should emphasize these more clearly. Specifically, the analysis reveals how few cites (if any!) are listed in resources addressing Indigenous issues. The authors discuss explanations in lines 421-430. While some of the rationales there are sensible, I want to offer a couple other perspectives. One is that despite there being less peer-reviewed work on on some of these subjects, there is some. TEK has a large literature, for example. Perhaps NEPA authors aren’t consulting the peer reviewed literature in Indigenous studies and other areas. This would be consistent with the evidence I’ve seen that social sciences are underrepresented in NEPA analyses. Another perspective is that the lack of citations reflect a deep engagement with Indigenous people. Although the NEPA analyses pay considerable attention to the issue, the analysis of resource impacts may be more performative.

15) I would like to know a little bit more about the BLM offices in Colorado, and the extent to which we can learn how much they differ in context and citation metrics.

Minor:

1) Change parentheses in line 8 to “The Biden administration…”

2) Line 112, what is “clear science use”?

References

Jacob D. Hileman, Mario Angst, Tyler A. Scott, Emma Sundström. 2021. Recycled text and risk communication in natural gas pipeline environmental impact assessments. Energy Policy 156.

Reviewer #2: This is an interesting paper on an important topic – the use of science in policy decisions. It analyses the citation of scientific and data documents in environmental assessments in a particular policy context and in a particular location.

The work appears to have been thorough and competent. The paper is well written and presented.

Although the analysis is simple in it mathematical/statistical approach, it is appropriate for the research question being addressed.

The paper fills a research gap and potentially establishes a foundation for future research in this area.

I found only one minor typo. Page 9, line 197, change “use” to “used”.

6. PLOS authors have the option to publish the peer review history of their article (what does this mean? ). If published, this will include your full peer review and any attached files.

**Do you want your identity to be public for this peer review?** For information about this choice, including consent withdrawal, please see our Privacy Policy .

Reviewer #1: No

Reviewer #2: **Yes: ** David Pannell

---

## [Author Response · Author response to Decision Letter 1]

27 Nov 2024

Thanks for the constructive review! Please see the word file for a detailed response to reviewers.

---

## [Editor Report · Decision Letter 1]

5 Dec 2024

Exploring the science and data foundation for Federal public lands decisions

PONE-D-24-09193R1

Dear Dr. Foster,

We’re pleased to inform you that your manuscript has been judged scientifically suitable for publication and will be formally accepted for publication once it meets all outstanding technical requirements.

Kind regards,

Frank H. Koch, PhD

Academic Editor

PLOS ONE

Additional Editor Comments (optional):

I commend the authors for their thoughtful responses to the reviewers' comments. The revised manuscript reads well and should find an appreciative audience. It's definitely suitable for publication.
---

## [Editor Report · Acceptance letter]

PONE-D-24-09193R1

PLOS ONE

Dear Dr. Foster,

I'm pleased to inform you that your manuscript has been deemed suitable for publication in PLOS ONE. Congratulations! Your manuscript is now being handed over to our production team.

Kind regards,

on behalf of

Dr. Frank H. Koch

Academic Editor

PLOS ONE